# Generation and Characterization of a Novel Angelman Syndrome Mouse Model with a Full Deletion of the *Ube3a* Gene

**DOI:** 10.3390/cells11182815

**Published:** 2022-09-09

**Authors:** Linn Amanda Syding, Agnieszka Kubik-Zahorodna, Petr Nickl, Vendula Novosadova, Jana Kopkanova, Petr Kasparek, Jan Prochazka, Radislav Sedlacek

**Affiliations:** 1Laboratory of Transgenic Models of Diseases, Institute of Molecular Genetics of the Czech Academy of Sciences, 252 50 Vestec, Czech Republic; 2Czech Centre for Phenogenomics, Institute of Molecular Genetics of the Czech Academy of Sciences, 252 50 Vestec, Czech Republic

**Keywords:** Angelman syndrome, UBE3A, mouse model, neurodevelopmental disease, autism spectrum disorder

## Abstract

Angelman syndrome (AS) is a neurodevelopmental disorder caused by deficits in maternally inherited *UBE3A*. The disease is characterized by intellectual disability, impaired motor skills, and behavioral deficits, including increased anxiety and autism spectrum disorder features. The mouse models used so far in AS research recapitulate most of the cardinal AS characteristics. However, they do not mimic the situation found in the majority of AS patients who have a large deletion spanning 4–6 Mb. There is also a large variability in phenotypes reported in the available models, which altogether limits development of therapeutics. Therefore, we have generated a mouse model in which the *Ube3a* gene is deleted entirely from the 5′ UTR to the 3′ UTR of mouse *Ube3a* isoform 2, resulting in a deletion of 76 kb. To investigate its phenotypic suitability as a model for AS, we employed a battery of behavioral tests directed to reveal AS pathology and to find out whether this model better mirrors AS development compared to other available models. We found that the maternally inherited Ube3a-deficient line exhibits robust motor dysfunction, as seen in the rotarod and DigiGait tests, and displays abnormalities in additional behavioral paradigms, including reduced nest building and hypoactivity, although no apparent cognitive phenotype was observed in the Barnes maze and novel object recognition tests. The AS mice did, however, underperform in more complex cognition tasks, such as place reversal in the IntelliCage system, and exhibited a different circadian rhythm activity pattern. We show that the novel UBE3A-deficient model, based on a whole-gene deletion, is suitable for AS research, as it recapitulates important phenotypes characteristic of AS. This new mouse model provides complementary possibilities to study the *Ube3a* gene and its function in health and disease as well as possible therapeutic interventions to restore function.

## 1. Introduction

Angelman syndrome (AS) is a rare congenital neurodevelopmental disease affecting one in 10,000 to 40,000 births. The clinical manifestations in individual patients differ considerably. However, frequent phenotypes include a jerky ataxic gait, little to non-existent speech, profound mental retardation, sleep disturbances, and hyperactivity early in childhood [1,2]. Seizures occur in approximately 80% of patients and are often considered a characteristic of AS [3]. The disease stems from the imprinted 15q11.2–13q locus, meaning that it exhibits parent of origin specific expression of certain genes, hence defying classic Mendelian genetics. The main causative gene of AS is *UBE3A,* which encodes a ubiquitin E3 ligase. The gene is paternally imprinted in neurons, only allowing for maternal expression. However, it exhibits bi-allelic expression in non-CNS tissue. The paternal allele is silenced by a large antisense transcript, commonly referred to as UBE3A-ATS, by a mechanism still being discussed [4]. The differential expression of genes in the locus are under the overall control of an imprint control region (ICR) acting in cis [4].

There are four main genetic etiologies causing AS, namely, (i) the de novo interstitial deletion of 15q11.2–13q on the maternally inherited chromosome [5]; (ii) paternal uniparental disomy [6]; (iii) imprinting defects due to aberrant epigenetic modification of the AS imprinting control region necessary for correct regulation [7]; and (iv) point mutations on the maternally inherited *UBE3A* gene. As mutations of this gene alone can cause AS, albeit with a milder phenotype, it is thus labeled as the AS gene [8,9]. The most common genetic cause for AS is the large deletion of the maternal 15q11.2–13q, which is found in approximately 70% of patients [10].

Although the occurrence of AS is equally distributed in both sexes, there is very little reported on the sex-dependent differences in humans [11,12]. However, few studies have reported sex-dependent differences in weight, anxiety, and cognitive and sensory phenotypes in AS mice [12,13,14,15]. In a study published by Koyavski et al. (2019) [12], they reported multiple genes exhibiting opposite sex-dependent transcriptional profiles between wild-type and AS mice. Many of them were linked to sensory phenotypes and several were estrogen-dependent, clearly showing the importance of sex on behavioral assessment [12].

For AS research, in terms of construct validity, the mouse organism serves as a useful and relevant model, as it has a syntenic region to the 15q11.2–13q locus on its chromosome 7, although with inverted orientation [16]. As for phenotypic similarity, the model needs to recapitulate robust phenotypes representative of AS to be useful for translational research.

Various mouse models mimicking AS have been generated and characterized, including deletions and substitutions of *Ube3a*, conditional *Ube3a* alleles, inducible *Ube3a* isoforms, *Ube3a* fusion to reporters, and large deletions encompassing multiple genes including *Ube3a* (Table 1) [17,18,19]. Models with intragenic modifications have been used to uncover the functions of the gene and its specific domains or isoforms (Table 1) [18,20,21,22]. Conditioned models allow for the cell-type-, tissue-, or developmental-stage-specific knockout of *Ube3a* and show the importance of spatiotemporal *Ube3a* expression [23,24,25]. Moreover, the role of individual *Ube3a* isoforms can be studied with inducible models such as Tg(tetO-Ube3a*2)^884Svd^ or Tg(tetO-Ube3a*1)^1Svd^ (Table 1). The models with tagged *Ube3a* were shown to be invaluable for the research, especially in deciphering the regulation, dynamics, and distribution of *Ube3a* expression in the studied models [26,27,28,29].

The most used mouse model in Angelman research is the Ube3a^tm1Alb1^ generated by Beaudet and colleagues [20]. This model, based on a deletion of exon 5, referring to isoform 2, with a truncated and non-functional UBE3A, has proven to have a high face validity in phenotypes linked to motor skills, vocalization, seizure susceptibility, and behavioral patterns such as repetitive behavior but does not recapitulate severe cognitive deficits normally assessed by fear conditioning and water maze or similar test. Instead, the cognitive malfunction presents as mild or absent, depending on the strain and experimenter [35]. In a recent study, it was demonstrated that AS mice on the C57BL/6J background displayed behavioral impairments and the B6 AS mice differed in the cortical EEG power spectrum compared to WT mice during the light cycle, with significant increases in delta and theta power. The AS mice on the 129 background exhibited poor performance on the wire hang and contextual fear conditioning. The Ube3a^tmAlb1^ animals also had a lower seizure threshold, both chemically and audiogenically induced, in comparison to wild-type animals [36]. The Ube3a^tmAlb1^ F1 hybrid mice on the 129 and C57BL/6 backgrounds appeared to have milder phenotypes altogether [36]. The reported variations have demanded a further improved AS model with higher translational value.

The majority of AS patients harbor a large deletion including the entire *UBE3A* gene. However, the mouse model used in AS research is limited to a 3kb deletion [8]. We are describing a new mouse model in which the entire *Ube3a* gene is deleted, encompassing both the coding and non-coding elements of the gene, to obtain a higher similarity to the patients’ situation. Furthermore, this new model was generated as part of the International Mouse Phenotyping Consortium (IMPC, www.mousephenotype.org accessed: 25 March 2022), which is aiming to produce and phenotype a knockout mouse model for every protein-coding gene on a similar genetic background and utilizing shared standard operating procedures for phenotyping. The model was assigned the strain name C57BL/6NCrl-Ube3a<em1(IMPC)Ccpcz>/Ph, further referred to as Ube3a^Genedel^. We characterized the Ube3a^Genedel^ model using a battery of phenotyping tests, adapted from the Sonzogni et al. (2018) [15], describing the AS pathology. These include tests connected to motor performance, stereotypic behavior, anxiety, and seizure susceptibility [15]. We adopted this test battery to either prove or disprove its utility in AS research. We found impaired motor skills, robust behavioral differences, changed circadian activity, and underperformance in memory flexibility.

## 2. Materials and Methods

### 2.1. Generation of the Ube3a^genedel^ Model

The Ube3a^Genedel^ mouse (C57BL/6NCrl-Ube3a<em1(IMPC)Ccpcz>/Ph) was generated on a C57BL/6N background (Charles River Laboratories) by targeting the 5′UTR and the 3′UTR of isoform 2 flanking the *Ube3a* gene (transcript Ube3a-203 ENSMUST00000200758.4) for a gene deletion by using the CRISPR/Cas9 technique. The guide RNAs (gRNAs) of highest score and specificity were designed using http://crispor.tefor.net/, accessed on 31 July 2022. The following guides were selected: Ube3a 5′UTR gRNAs, U5-1: 5′-CGCGGGTCCCGCATGAGACC-3′ and U5-2: 5′-GCGCTTCAGCGCCGACTTCA-3′ and Ube3a 3′UTR gRNAs, U3-1: 5′-CCTTGCGAGAATAGTTTCGT-3′ and U3-2: 5′-CTGTCCTTTCATATACTAAG-3′. The gRNAs were assembled into a ribonucleoprotein (RNP) complex with Cas9 protein (1081058, 1072532, Integrated DNA technologies), electroporated into 1-cell zygotes, and transferred into pseudopregnant foster mice. Putative founders were analyzed by PCR and sequencing. A founder harboring a 76,225 bp deletion spanning the entire *Ube3a* gene was chosen for subsequent breeding. Genotyping was performed by junction PCR with 5′UTR forward (5F) 5′-CAGTCTCAAGATGGCGACGA-3′, 5′UTR reverse (5R) 5′-CAATCCACCCCCAATACCCC-3′, and 3′UTR reverse (3R) 5′-GCTACCATTATCCCCTGCCAA-3′. The 5F and 5R products are 1183 bp, seen in all wild-types and AS mice. The 5F and 3R, only present in AS animals, have a size of 1602 bp, otherwise located ~77 kb apart in wild-type animals.

### 2.2. Mouse Husbandry, Breeding, and Experimental Cohorts

All animals and experiments used in this study were ethically reviewed and performed in accordance with European directive 2010/63/EU and were approved by the Czech Central Commission for Animal Welfare. Mice were housed in individually ventilated cages (Techniplast), genotyped at 14–21 days of age, and regenotyped after a battery of tests. All animals were kept at 22 ± 2 °C with a 12 h dark and light cycle and were tested during the light period and provided with mouse chow (Altramion) and water ad libitum. The transgenic mice were maintained on a *Ube3a*^+/−^ zygosity, and animals for experiments were generated by crossing female *Ube3a*^+/−^ animals with wild-type males to attain *Ube3a*^−/+^ progeny and control littermates. Weight measurements started at four weeks and were repeated once per week until 15 weeks (WT *n* = 50; AS *n* = 26). All testing was performed during the light phase of the day on 9–18-week-old male and female mice. The testing was conducted on three animal cohorts consisting of an approximate 1:1 ratio between female and male mice for both genotypes (Table 2). Male and female mice were tested separately on different days. The tests in each cohort were conducted as follows: cohort 1 was tested with the rotarod, DigiGait, and tail suspension tests and nest building, cohort 2 was tested with the elevated plus maze, open field, novel object recognition test, and Barnes maze; and cohort 3 was tested with IntelliCage. Testing arenas, mazes, objects, and animal enclosures were thoroughly cleaned with 75% ethanol and then dried to remove olfactory traces before the first tested animal and between the tested animals.

### 2.3. Western Blotting

Tissues were dissected, snap-frozen in liquid nitrogen, and lysed in RIPA buffer (0.05 M Tris-HCl, pH 8, 0.15 M NaCl, 0.5% deoxycholic acid, 1% NP-40, and 0.1% sodium dodecyl sulfate (SDS)), cOmplete™, EDTA-free Protease Inhibitor Cocktail (Roche, 5056489001), and PhosSTOP™ phosphatase inhibitor tablets (Roche, 4906845001, Darmstadt, Germany). Lysates were sonicated and cleared by centrifugation; the protein concentration was determined using the Pierce™ BCA Protein Assay Kit (Thermo Scientific, 23225, Rockford, IL, USA). Equal amounts of protein (20 μg) were analyzed by Western blotting, where anti-Ube3a (1:2000, BD Biosciences, 611416, Brno, Czech Republic) and anti-B-actin (1:5000, Sigma-Aldrich, A2228, Darmstadt, Germany) were used as primary antibodies. Blots were washed with PBS-T, and detection was performed with SuperSignal™ West Pico PLUS Chemiluminescent Substrate (ThermoScientific, 34579, Rockford, IL, USA). The Western blots were quantified using ImageJ analysis, and the sample size was *n* = 8 with a 1:1 ratio for the sexes.

### 2.4. Rotarod

The rotarod system was used to assess the sense of balance, motor learning, and motor coordination in the animals [37]. Three trials per day for each mouse were recorded on a rod with an accelerating speed of rotation (4–40 rpm/5 min, RotaRod, TSE Systems GmhB, Berlin, Germany) during five consecutive days. The average latency to fall was determined from the three trials with 15 min intratrial intervals (ITI).

### 2.5. Gait Analysis

Each animal was placed on a motorized treadmill with a belt speed of 20 cm/s (DigiGait, Mouse Specifics, Ins., Framingham, MA, USA). A minimum of five fluent strides were recorded by a camera positioned below the belt to focus on the ventral view of the animal. The DigiGait software (Mouse Specifics, Ins., Framingham, MA, USA), which recognizes and evaluates over 50 gait indices describing walk quantitative metrics, kinematics, and animal posture, automatically analyzed the recordings [38].

### 2.6. Tail Suspension Test

The tail suspension test (TST) was used to assess tendencies for depressive-like behavior [39]. The tested animal was attached to an apparatus hook and left suspended for six minutes. The immobility time, energy, and power in motion were analyzed using BIO-TST 4.0.2.1 (Bioseb, Vitrolles, France) software.

### 2.7. Open Field

The activity of the animals in a novel environment and the displayed level of anxiety were evaluated in open field (OF) tests as previously described [40]. The area of the open field was a square of 42 × 42 cm that was uniformly illuminated with a light intensity of 200 lux at the center of the field. The testing arena was virtually divided into periphery and center zones, where the center zone constituted 38% of the whole arena. Each mouse was placed in the corner of the arena for a 10 min period of free maze exploration. The time spent in each zone, the distance travelled, and other indices were automatically computed based on video recordings (Viewer, Biobserve GmbH, Bonn, Germany).

### 2.8. Elevated plus Maze

The elevated plus maze (EPM) apparatus consisted of two closed and two open elevated arms, with a light intensity of 60 lux at the center of the maze [41]. The subject animal was placed at the center and was left to explore the EPM for five minutes. The total time spent in the open and closed arms and the center of the EPM were tracked and evaluated automatically (Viewer software, Biobserve GmbH, Bonn, Germany).

### 2.9. Novel Object Recognition

The novel object recognition test (NOR) was used to evaluate animal exploration of a novel object as a measure of working memory and attention [42]. The test, fully automated and based on a video tracking system (Viewer, Biobserve GmbH, Bonn, Germany), was preceded by two days of 10 min-long habituation to the testing maze. Familiarization with two identical objects was performed on the following day. The probe trial with one familiar and one new object took place after a three-hour-long retention time. All trials were undertaken with 70 lux illumination at the center of the maze. The objects were placed 7.5 cm from the walls, alternating to different sides of the quadrant, and the subject was always placed on the opposite wall from the objects to counterbalance the relative position. Object visits, object exploration time, and the percentages of listed parameters were automatically computed for each object.

### 2.10. Nest Building

An evaluation of material usage for nest building was performed as previously described by Sonzogni et al. [15]. Mice were first separated to single cages for a week. Following accommodation to single housing, 12 g of FDA Nestlets (Datesand Ltd., Manchester, UK) were introduced to each cage for five consecutive days. The unused material in the nest was weighed at the same time each day.

### 2.11. Barnes Maze

Spatial learning and memory were analyzed according to a previously published protocol [43]. We used a random pattern of holes to prevent mice from using a serial strategy to find the escape box [44]. Except for the first habituation day, where a low-light condition (70 lux) and no noise were used, the entire procedure was performed in high-aversive conditions, with an illumination of 200 lux and moderate background noise, to motivate the animals to search for the escape target box. During the two-day habituation phase, the mice were allowed to explore the maze for 7 min to find an escape box positioned in a different place each day. If the subject animal failed to find the escape box, it was gently guided in a glass container to the target and left there for 30 s. The next stage was a 5-day training session, during which four different escape box positions in the maze were used alternately as follows: The specific escape box position for each individual animal varied, but each individual animal was trained for the same escape box position during all 5 days of the training session. The mice were allowed to find the target hole during two 3 min-long trials with an approximately 2 h ITI on each day of the acquisition stage. The animals were tested in a probe trial 24 h after the last day of the acquisition. During the 1 min of the probe trial, no escape box was present in the maze. Video recordings were automatically analyzed by the Viewer software (Biobserve GmbH, Germany). Primary distance, latency, and errors to reach the target hole were used for further analysis. Additionally, the time and distance in the target quadrant were evaluated during the probe trial.

### 2.12. Testing in Home-Cage Environment IntelliCage

For the evaluation of circadian activity, we used the free adaptation protocol in the IntelliCage system (TSE Systems GmbH), which enables the testing of grouped animals in a home-cage environment [45], eliminating stressful conditions originating from animal handling during experimentation. Each IntelliCage is equipped with four operant corners with access to water. Each corner detects animal presence and identifies every individual with a radio-frequency identification (RFID) antenna that detects subcutaneously implanted transponder signals. The IntelliCage system collects information on entries and exits from the corner during a real time scale and records information on the frequency and duration of licks during each visit in every corner. The number of corner visits without licks was calculated to remove the effect of drinking motivation on general activity. Males and females were tested in separate cages. The mice were introduced to the IntelliCage for a free-adaptation regime for a week, and the animals had free access to water. The latencies to the first visit and lick were recorded to evaluate the animal novelty response. The temporal distribution of visits and their frequency were recorded to compare experimental group circadian activity. To assess spatial learning in place preference, each mouse was assigned to one rewarding (correct) corner to drink for five consecutive days. Upon visiting the correct corner, the mice were rewarded with 7 s of access to water. After the behavioral acquisition of the task, the learning flexibility was evaluated with a reversal place preference phase, where the correct corner was changed to the diagonally opposite one for each animal for five consecutive days.

### 2.13. Statistical Analyses

For the main genotype, sex, and interaction effects, a two-way ANOVA, with dependent measurements where needed, was used if the data distribution met the normality assumption. A robust ANOVA of aligned rank transformed data, with dependent measurement where needed, was used if the data distribution did not meet the normality assumption. Tukey’s post hoc test for normally distributed data or the Wilcoxon post hoc test for non-normal data distributions were used to assess differences within sex and genotype. Statistical analysis was performed in R version 4.1.2 (2021-11-01) with the stats version 4.1.2, ARTool version 0.11.1, and rstatix version 0.7.0 packages. All data are presented as means ± SD or medians with interquartile ranges.

## 3. Results

### 3.1. The New Ube3a^genedel^ Model Harbors a Large Deletion Spanning the Entire Ube3a Gene

The Ube3a^genedel^ model was generated using CRISPR/Cas9, resulting in a founder harboring a deletion of the gene from the 5′UTR to the 3′UTR (Figure 1A). Subsequent sequencing confirmed a deletion of 76,225 bp, mediated by corresponding gRNAs (Figure 1A); this founder was further selected for by the breeding of the F1 progeny.

Screening the putative founders with junction PCR revealed one founder with an appropriate amplification using the 5F, 5R, and 3R primer triplet (Figure 1B). Western blotting confirmed a near complete abolishment of protein expression in CNS tissue, with the limited detectable expression coming from non-neuronal cells. In the non-imprinted liver tissue, there was a decrease in UBE3A to half of that seen in WT animals (main genotype effect; *p* < 0.001; Figure 1C,D). The Ube3a^genedel^ mice and controls were weighed once per week from four weeks to 15 weeks of age, and both female and male Ube3a^genedel^ mice were significantly heavier than their litter mate controls (main genotype effect; *p* < 0.001; Figure 1E,F).

### 3.2. Gait Impairment and Impaired Motor Skills Are Recapitulated in The Novel Ube3a^genedel^ Mouse

Ataxia and severely impaired motor skills are consistently present in individuals with AS, manifesting with an ataxic gait with tremor. Furthermore, this phenotype is one of the key clinical features for diagnosis [46]. We performed a detailed gait analysis using the DigiGait platform to assess the gait in the animals. This system allows for both spatial and temporal analyses of gait parameters. We found that the Ube3a^genedel^ mice exhibited prolonged durations in parameters such as swing, stance, stride, and propel (main genotype effect; F = 48.131; F = 54.346; F = 94.802; F = 20.098; *p* < 0.001 for all parameters; Figure 2A–E). Other characteristics, such as MAX dA/dt (maximal rate of change of paw area in contact with the treadmill belt during the braking phase), paw area at peak stance, and stride length, were increased (main genotype effect; F = 26.929; F = 33.725; *p* < 0.001 for all parameters; Figure 2F,G).

The motor coordination and balance of the mice were evaluated on the rotarod apparatus. We found that the Ube3a^genedel^ mice had a significantly shorter latency to fall, which further confirmed the observation that the mice have motor skill dysfunction (main genotype effect; F = 54.111; *p* < 0.001; Figure 2H,I). The weights of the mice were plotted against the latency to fall to investigate any putative weight bias. However, weight was not a significant factor (factor of weight; *p* > 0.05; Appendix A).

### 3.3. The Ube3a^genedel^ Model Exhibits Robust Behavioral Impairment

To assess general activity and anxiety, we used open field and elevated plus maze tests. In the open field test, male Ube3a^genedel^ mice appeared to be hypoactive, as the distance walked and average speed were decreased, whereas the resting time was increased (main genotype effect; F = 9.233; F = 9.268; F = 10.033; *p* < 0.01 for each parameter; sex/genotype interaction *p* > 0.05; Figure 3A–C); females exhibited similar tendencies, but the results were not significant. However, no significant differences in the number of entries into the anxiogenic center nor the time spent there were observed, which indicates that there is no general anxiety-like phenotype in Ube3a^genedel^ mice (main genotype effect *p* > 0.05; F = 3.34; F = 0.00; Figure 3D,E). The observation was confirmed by the results from the elevated plus maze, where mice of both genotypes spent comparable time in the anxiogenic open arms (main genotype effect; F = 2.88759; *p* > 0.05; sex/genotype interaction *p* > 0.05; Figure 3F–H).

The mice were further tested for other behavioral phenotypes, such as their innate instinct to create a nest via the nest building test [47]. Here, we found that Ube3a^genedel^ mice underperformed in the task, as commonly occurs in mice with neurodegenerative disease [47] (main genotype effect; F = 4.86599; *p* < 0.05; sex/genotype interaction *p* > 0.05; Figure 4A,B). The results from TST showed that Ube3a^genedel^ mice spent a significantly longer time immobile compared to the WT controls (main genotype effect; F = 20.326; *p* < 0.001 sex/genotype interaction *p* > 0.05; Figure 4C) which is a characteristic of depressive-like behavior.

### 3.4. Memory and Learning Were Not Impaired in Ube3a^genedel^ Mice in Barnes Maze and Novel Object Recognition Tests

Severe cognitive disabilities are a key characteristic of AS, and we aimed to assess the cognitive phenotype in the Ube3a^genedel^ model using the Barnes maze and NOR tests. The learning curve, measured by the latency and primary distance needed to find an escape box during the Barnes maze learning phase, did not differ between genotypes (main genotype effect *p* > 0.05; F = 0.478; sex/genotype interaction *p* > 0.05; Figure 5A,B). The memory of the animals also did not differ in the probe trial, where the time spent and distance walked in the target quadrant were estimated (main genotype effect *p* > 0.05; F = 2.566; sex/genotype interaction *p* > 0.05; Figure 5C).

When subjecting the mice to the NOR test, we found no significant difference in the percentage of the duration spent with the new object between the genotypes or sexes (main genotype effect; F = 0.088525; *p* > 0.05; sex/genotype interaction *p* > 0.05; Figure 5D). We conclude, based on these experiments, that the animals do not exhibit cognitive impairments in easier tasks.

### 3.5. Home-Cage Circadian Activity, Response to Novelty, and Performance in Place Reversal Tasks Differ in Ube3a^genedel^ Animals

The circadian activity, drinking behavior, and memory of the mice were evaluated using the IntelliCage paradigm. Male Ube3a^genedel^ mice exhibited significant increases in latency to the first corner visit and the first lick after introducing animals into the IntelliCage (main genotype *p* < 0.001; F = 20.139, F = 4.561; genotype/sex interaction *p* < 0.05; Figure 6A,B). A post hoc analysis showed that only the Ube3a^genedel^ males exhibited increased latency in the tested parameters (Figure 6A,B).

The number of corner visits was recorded and displayed over the course of the day (Figure 6C,D). There were no significant differences in total visits (main genotype effect *p* > 0.05; Appendix A). However, there were differences in activity depending on the phase of the day (main genotype effect; *p* < 0.001; F = 1.35; genotype/time interaction *p* < 0.05; Figure 6C,D). A subsequent post hoc analysis revealed that the Ube3a^genedel^ mice’s decrease in corner visits was pronounced during the light phase of the day (Figure 6C,D). When visits with licks were included, the significant difference disappeared (main genotype effect *p* > 0.05; Appendix A).

The number of licks was increased in the Ube3a^genedel^ mice during a 24 h period (main genotype effect; F = 10.52; *p* < 0.01; sex/genotype interaction *p* > 0.05; Figure 6E,F). A subsequent post hoc analysis showed that this was pronounced during the dark phase, but the opposite tendency was seen during the light phase. However, the overall number of licks during a 7-day period showed a clear increase in licks in Ube3a^genedel^ animals (main genotype effect *p* < 0.001; Appendix A). An additional analysis showed that AS animals made more licks per visit (main genotype effect *p* < 0.001; sex; genotype interaction *p* > 0.05; Appendix A).

We found no significant difference between Ube3a^genedel^ and WT mice in place preference phase (main genotype effect *p* > 0.05; F = 0.01; Figure 6G,H).

Place preference reversal learning showed that Ube3a^genedel^ animals significantly underperformed, exhibiting a lower percentage of correct visits to the rewarding corner (main genotype effect *p* < 0.05; F = 5.04; sex/genotype interaction *p* > 0.05; F = 0.10; Figure 6I,J).

## 4. Discussion

We employed the CRISPR/Cas9 technology to produce a new AS model, deleting *Ube3a* from the 5′UTR to the 3′UTR. In regard to the isoform in lysates from the Ube3a^genedel^ mice, a Western blot analysis of several regions in the CNS showed a near complete reduction in the UBE3A protein, with parts of it remaining due to the paternal contribution from non-neuronal cells [48]. The liver, a tissue exhibiting bi-allelic expression of UBE3A, was also analyzed, showing a near 50% reduction in the protein.

The rationale to produce a mouse model mimicking AS, in addition to the already existing and well-characterized models, was to generate a model harboring a deletion encompassing the entire *UBE3A* gene, as ~70% of AS patients have a large deletion of the locus [20]. This could complement other AS models and reveal additional information about gene importance in addition to the most studied model generated by the Beaudet lab [20].

To evaluate our new model in terms of its phenotypic suitability for AS research, we subjected it to a battery of tests aimed at the characterization of AS phenotypes. We used the behavioral test battery put forth by the Elgersma lab, with some adaptation [15]. Ataxia and severely impaired motor skills are consistently present in AS patients, manifesting in an ataxic gait with tremor. This phenotype, one of the key clinical features for diagnosis [49], was demonstrated in the Ube3a^tmAlb1^ model [50]. For instance, the mice performed poorly on the accelerating rotarod and exhibited a distorted gait with wider, longer, and fewer steps compared to wild-type littermates [36,51].

We evaluated motor skills with the rotarod test and with the detailed gait analysis in DigiGait. It was revealed that the Ube3a^genedel^ mice significantly underperformed in the rotarod task, with a shorter latency to fall. This was true for both female and male Ube3a^genedel^ mice, although the increased weight in Ube3a^genedel^ mice could be a confounding factor. However, a linear regression analysis revealed no significant weight/latency correlation, thereby excluding the influence of animal weight on rotarod performance (Appendix A). Our data also corresponded to observations made in another study, clearly indicating that the increased weight in AS animals is not a reason for underperformance [15]. In addition, we performed a detailed gait analysis using the DigiGait system to evaluate each of the four limbs, ultimately defining the posture and kinematics and allowing for the extrapolation of strength, coordination, and balance in the animals. We assessed the duration of common gait characteristics such as swing, stance, propel, and stride. We found an increase in the duration of all four parameters and longer strides. An increase in the propel duration, defined as the time spent in the air between steps, indicates reduced strength and control of movement [52]. Additionally, we collected data on the deceleration and the paw area at peak stance. We found that the Ube3a^genedel^ mice exhibited increased deceleration, which has been reported to be an indicator of reduced muscle strength [53]. A larger contact area of the paws at peak stance may act to stabilize the posture and balance [54].

Based on our data collected from the rotarod and DigiGait, we conclude that the novel Ube3a^genedel^ mice recapitulate the motor deficits seen in AS patients and could be of translational value for therapeutics aimed towards improving the motor phenotype.

We set out to evaluate whether our new Ube3a^genedel^ model exhibits any behavioral impairments, which are a hallmark phenotype in AS patients [1], and found that both sexes of Ube3a^genedel^ mice were hypoactive in the open field test. This has been reported in multiple studies using other AS mouse models [15,35]. The Ube3a^genedel^ mice showed significantly increased resting time and decreased average speed and total distance walked. The detected hypoactive phenotype in our Ube3a^genedel^ model opposes what is found in AS patients. AS patients are frequently reported to exhibit hyperactivity during childhood. However, this was shown to decrease with age [55]. Regarding the evaluation of anxiety, there were no differences in time spent in the center or the number of entries into the center, suggesting that the mice are not more anxious than their WT littermates. We also subjected mice to the elevated plus maze test, where we did not observe any significant differences between the genotypes, again suggesting an absence of an anxiety-like phenotype. This is opposed to the findings in other studies, where the investigators have observed increased anxiety-like behavior when using the open field and elevated plus maze tests [15,24,36,56].

Our results are, however, in line with previously conducted studies on the Ube3a^tmAlb1^ model, where the mice clearly exhibit hypoactivity in open field tests. The reason for the observed hypoactivity could be the increased weight of the animals [15].

Another relevant neurobehavioral parameter is nest building, and the Ube3a^genedel^ mice exhibited a reduction in nest building activity, thus showing a sign of distress, a hallmark of neurodegenerative diseases [48]. Although we observed a main genotype effect, a subsequent post hoc analysis revealed that females were more affected, aligning with the fact that females are more likely to nest. Lastly, we evaluated the animals’ behavior in the tail suspension test, a commonly used tool to assess depressive-like behavior. Our Ube3a^genedel^ mice were significantly more immobile in this test, corresponding well with similar tests performed on the Ube3a^tmAlb1^ model [36], exhibiting no sex-dependent differences.

Regarding the cognitive functions, the Barnes maze test was used to assess any impairments in hippocampal-dependent learning, a phenotype presented in AS patients [44,57]. The results did not reveal any hippocampal-dependent cognitive malfunctions. Additionally, there were no significant differences in the NOR test assessing the short-term memory of animals. A possible reason for the lack of a cognitive phenotype is likely the age of the mice. In a study by Huang et al. (2013) [56], the authors showed that the AS mice from the B6 strain had deficits in spatial learning acquisition after 16 weeks but not at 8 weeks of age, showing that cognitive function becomes impaired with time [56]. As our cohort of mice were younger than 16 weeks at the time of testing in the Barnes maze and NOR test, this could potentially be the reason as to why we did not observe any cognitive dysfunction. We did, however, test the animals in place preference and reversal place preference learning paradigms, where mice needed to modify already learned behavior, observed using the IntelliCage setup at the age of 18 weeks. We observed a specific cognitive impairment of reversal learning in the Ube3a^genedel^ animals, but no impairment was seen in the initial place preference acquisition. A deficit in reversal learning has also been described in animal models of autism, where the Morris water maze or T-maze task were used [58,59,60]. Ube3a^genedel^ animals performed equally in reward-motivated place preference learning but showed deficiency in behavioral flexibility measured by reversal learning.

Lastly, we evaluated our Ube3a^genedel^ model for circadian activity and novelty response using the IntelliCage paradigm. The latency to the first visit of the corners and the latency to the first lick were increased in Ube3a^genedel^ males but not in females. The increased latency could be interpreted as a decreased exploratory drive and not necessarily due to increased anxiety-like behavior, which was strengthened by a detailed analysis of animal activity that revealed a marked attenuation of corner exploration during the light phase of the day [61]. Nonetheless, an important factor to take into account is the hypoactivity seen in our Ube3a^genedel^ mice in both sexes. However latency was only affected in males. The total number of corner visits over 24 h did not differ, whereas visits not motivated by drinking, visits without licks, were decreased during the light phase of the day. Ube3a^genedel^ animals drank more during the dark phase due to higher number of licks per visit. This could be due to the overall larger size of the Ube3a^genedel^ mice, but it could also be possible that the Ube3a^genedel^ mice are drawn to the water per se, as AS patients are characterized by a fascination with water [62]. However, we do not have the data to make such a statement.

The variation in animal ages needed for the different behavioral and cognitive paradigms is a clear barrier for testing new drugs or evaluating other therapeutic interventions in the Ube3a^tmAlb1^ mouse model, as cognitive deficits appear later on, while the reflex phenotype is seen only in juvenile mice [56].

As briefly mentioned in the introduction, it is important to take sex-dependent differences into account when evaluating behavioral and cognitive phenotypes in mice. Indeed, we did observe differences in nest building, where AS females used significantly less material for nest building than WT littermates, a difference that was not seen between WT and AS males. We also detected a more pronounced impairment in place reversal in AS females compared to WT than in males. The sex-dependent differences in the cognitive and behavioral aspects correspond well to findings reported by Koyavski and colleagues. They observed differences in neurobehavioral aspects where the AS mice either lacked differences between the sexes or showed opposed differences from WTs [12]. They did not observe any sex-dependent differences in motor phenotypes, again corresponding well with our results, as the AS mice of both sexes underperformed in the rotarod test and displayed distorted gait. Interestingly, they found sex-dependent differences in the transcriptome of AS mice as well, with several genes being estrogen-dependent [12]. Indeed, where the mice are in the estrous cycle plays a role in behavior and cognition, as the estrous-related hormones estrogen, progesterone, and their metabolites bind to steroid receptors in the brain, exerting influence on the mentioned parameters [63,64]. Unfortunately, we did not assess the estrous cycle in our females, which can be considered a limitation of the study method. There is limited information on sex-dependent differences in AS individuals, but the cycle in AS female subjects could possibly impact the assessment of the efficacy of therapeutic interventions and likely should be addressed.

In conclusion, we generated a new model in which the whole *Ube3a* gene was deleted. This target design differs from other existing models where smaller deletions are present, such as in the Ube3a^tmAlb1^ model harboring a 3kb deletion of exon 5 [20]. Genetic elements in the non-coding parts of the gene could possibly be present, thus potentially worsening the phenotype. We observed similar phenotypes in behavioral, motor, and cognitive tasks as in the Ube3a^tmAlb1^ model [36], although further studies, for instance with aged animals, are needed to confirm its usefulness. The motor impairment was particularly robust in our model and, based on previous work by Silva-Santos et al. (2015) [24], we now know that the developmental window for rescuing motor skills closes in adolescence, significantly later than the behavioral deficits that need *Ube3a* reinstatement during early development [24], which makes tests relying on the motor skills of the animal a good indicator of the success of therapeutics. This model provides several similarities to AS patients as well as several dissimilarities. We did not observe any cognitive deficits in simpler tasks, such as the Barnes maze and NOR tests, but that could be due to the age of the animals [56]. The memory flexibility was, however, affected in the Ube3a^genedel^ animals. There are other phenotypes associated with AS that were not evaluated in this study, such as electrophysiological phenotypes, abnormal EEG, and autistic behavior. The investigation of these phenotypes could lay a foundation for future publications. Nevertheless, this model can be rendered suitable for AS research and the potential testing of therapeutic interventions.

## Figures and Tables

**Figure 1 cells-11-02815-f001:**
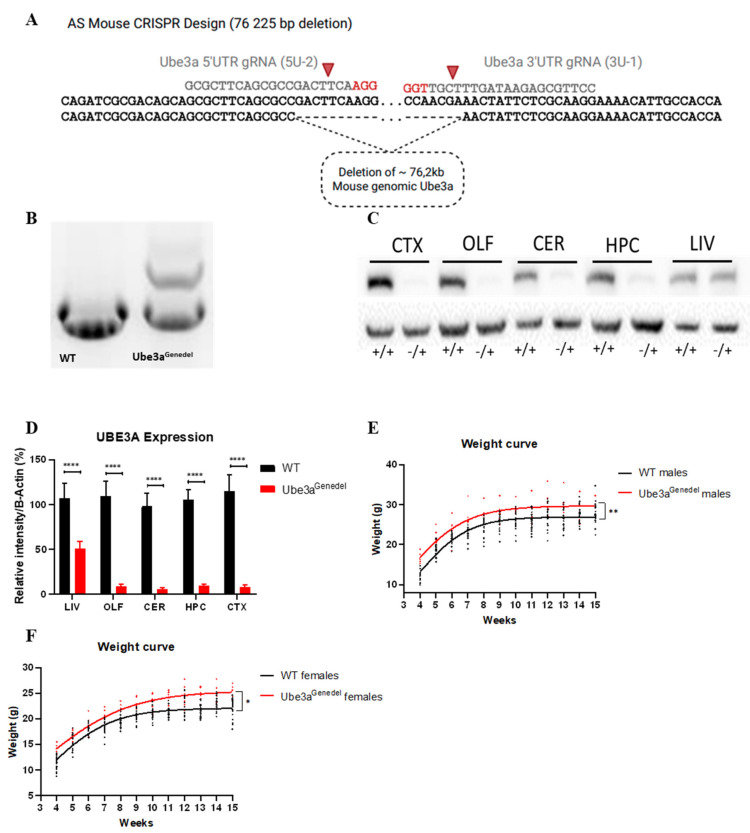
The Ube3a^Genedel^ mouse model. (**A**) Schematic of the Ube3a^Genedel^ deletion, a 76,225 bp deletion range from 58,878,821 to 58,955,045, *Mus musculus* strain C57BL/6J chromosome 7, GRCm39 (NC_000073.7). The gRNA generating the cut are indicated. (**B**) Representative amplification of WT control and Ube3a^Genedel^ DNA, with the WT showing one band of 1183 bp and the Ube3a^genedel^ showing an additional at 1602 bp. (**C**) A representative Western blot showing a near complete reduction in UBE3A in several brain regions and a decrease to half in liver tissue. (**D**) A quantification of the UBE3A expression from Western blot; two-way ANOVA main genotype effect *p* < 0. CTX: cortex, OLF: olfactory bulb, CER: cerebellum, HPC: hippocampus, LIV: liver. (**E**,**F**) Graph of weight recordings from 4 weeks to 15 weeks. All data represent means ± SD. Significant effects of genotype are indicated as * *p* < 0.05, ** *p* < 0.01 and **** *p* < 0.0001 for genotype significance.

**Figure 2 cells-11-02815-f002:**
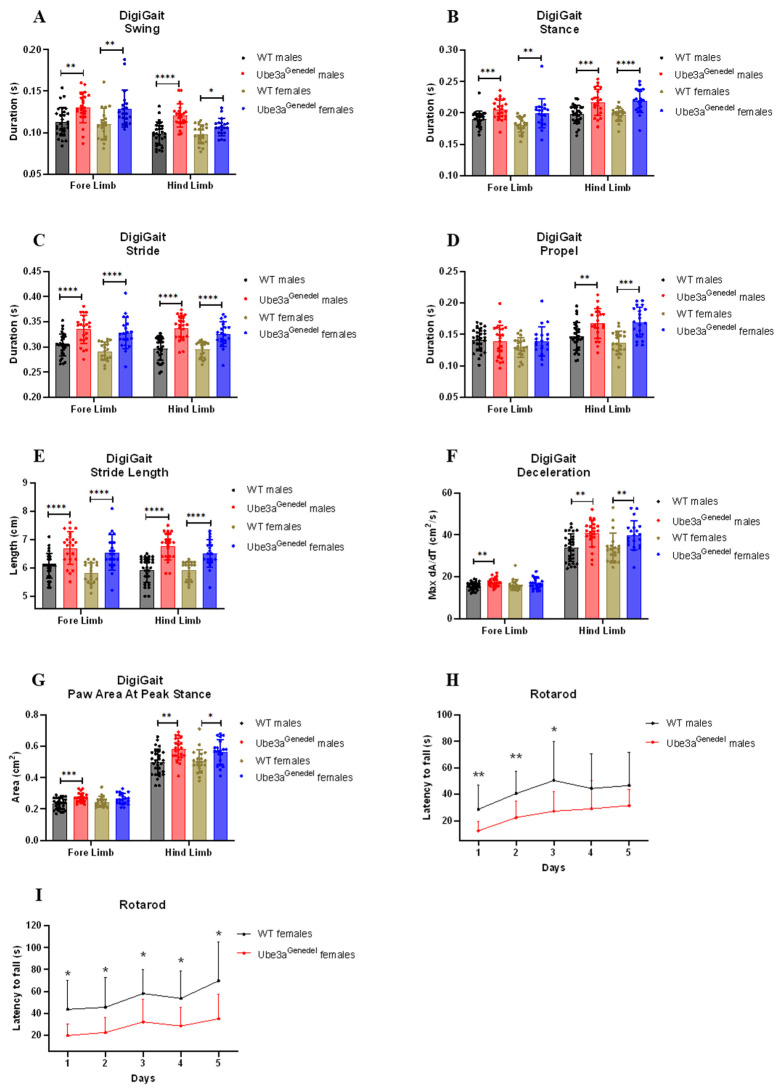
Results of gait analysis and rotarod. (**A**–**G**) Gait parameters were measured with the DigiGait platform. The data show an increase in the duration of swing, stance, stride, and propel in Ube3a^genedel^ animals. Stride length, MAX dA/dt, and the paw area at peak stance were also increased in Ube3a^genedel^ animals in both sexes; two-way ANOVA: genotype main effect, *p* < 0. (**H**,**I**) Ube3a^genedel^ mice had significantly shorter latency to fall; two-way ANOVA with dependent measurements genotype main effect: *p* < 0. All data represent means ± SD. Significant effects of genotype are indicated as * *p* < 0.05, ** *p* < 0.01, *** *p* < 0.001, and **** *p* < 0.0001 for genotype significance.

**Figure 3 cells-11-02815-f003:**
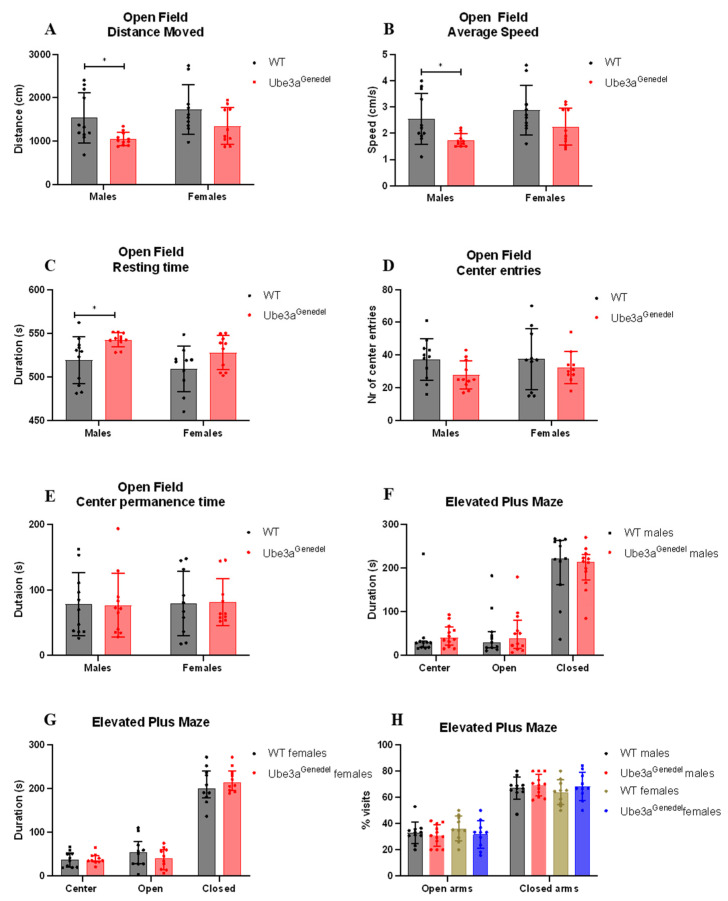
Activity and assessment of general anxiety with open field and elevated plus maze. (**A**–**E**) The performance in the open field for the parameters total distance moved, average speed, resting time, center entries, and center permanence time. The Ube3a^genedel^ mice exhibited increased resting time, while the total distance moved and average speed were decreased; two-way ANOVA with dependent measurements genotype main effect, *p* < 0.01, depicted with means ± SD. (**F**–**H**) Time spent in the center, open, and closed configuration of the EPM, showing no significant alteration in duration or percentage between the genotypes; robust ANOVA of aligned rank transformed data with dependent measurements genotype main effect, *p* > 0.05, depicted with medians and interquartile ranges. Significant effects of genotype are indicated as * *p* < 0.05 for genotype significance.

**Figure 4 cells-11-02815-f004:**
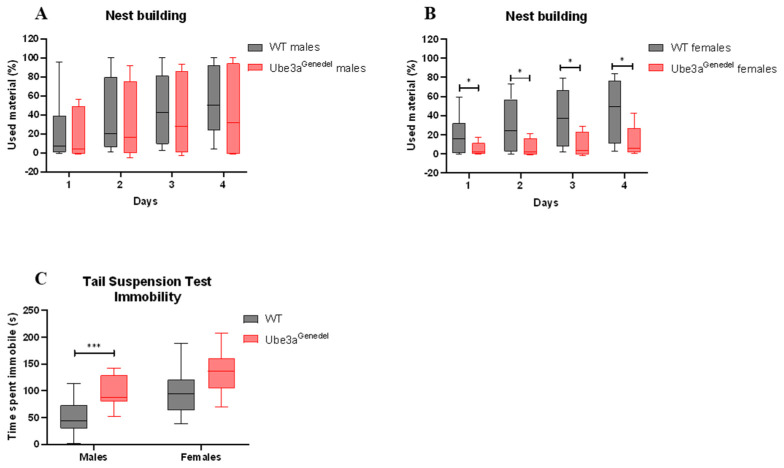
Results from nest building and tail suspension tests. (**A**,**B**) The Ube3a^genedel^ mice used significantly less material for nesting; robust ANOVA of aligned rank transformed data with dependent measurements, genotype main effect, *p* < 0.05, depicted with medians and interquartile ranges. (**C**) Ube3a^genedel^ mice spent significantly more time immobile during the last four minutes of the test; two-way ANOVA with dependent measurements, genotype main effect, *p* < 0.001, depicted with means ± SD. Significant effects of genotype are indicated as * *p* < 0.05 and *** *p* < 0.001 for genotype significance.

**Figure 5 cells-11-02815-f005:**
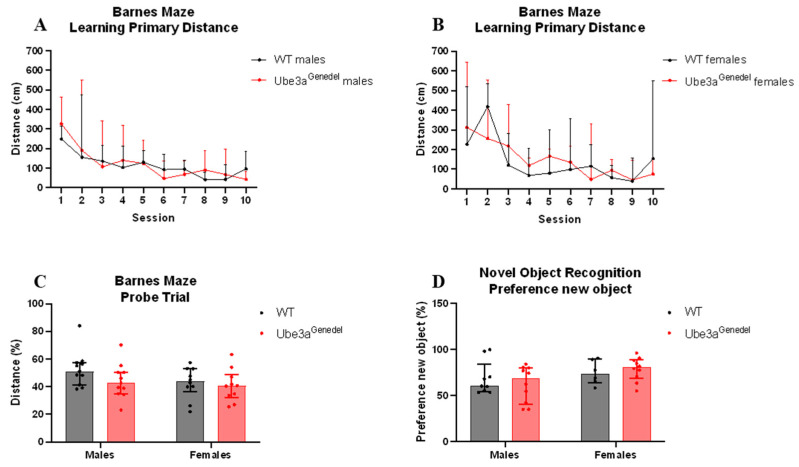
Learning and memory assessment. (**A**–**C**) The durations of the learning phase and probe trial were not significantly altered between genotypes; robust ANOVA of aligned rank transformed data with dependent measurement, *p* > 0. (**D**) No significant differences in the percentage of time spent with a new object between the genotypes, robust ANOVA of aligned rank transformed data with dependent measurement main effect, *p* > 0. All data represent medians with interquartile ranges.

**Figure 6 cells-11-02815-f006:**
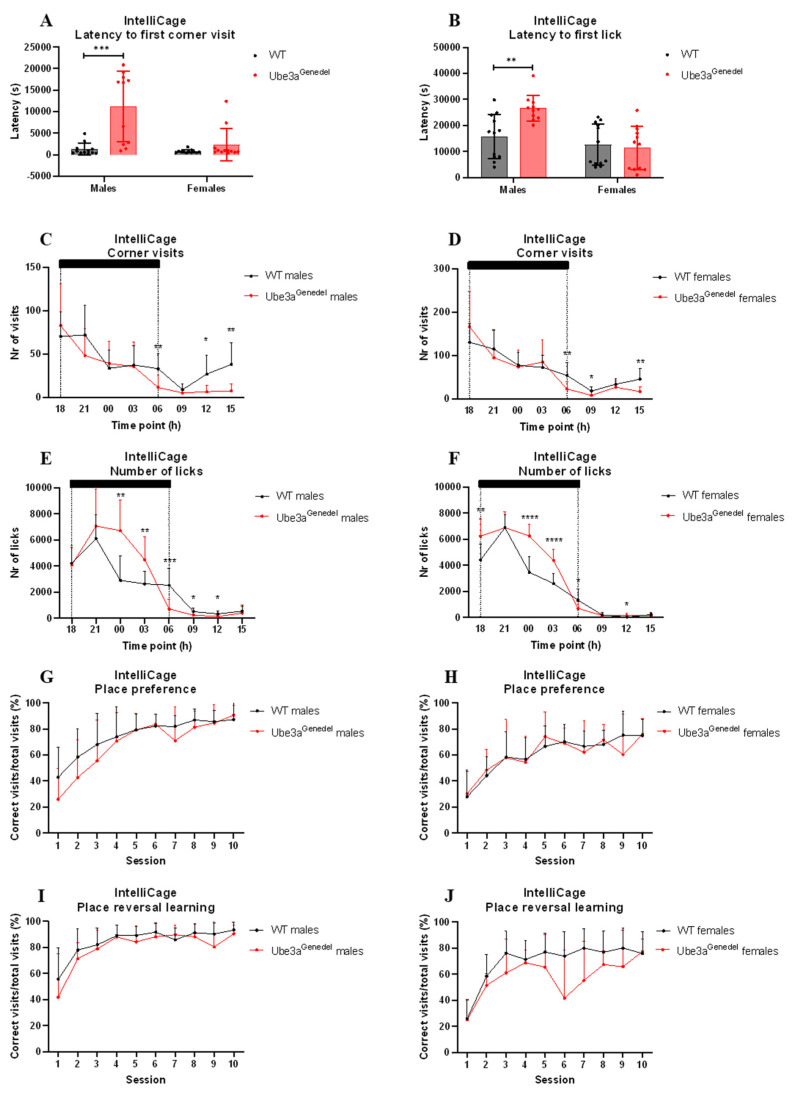
Results from IntelliCage measurements. (**A**,**B**) Ube3a^genedel^ male mice exhibited increased latency to first corner visit and first lick over 7 days. (**C**,**D**) Ube3a^genedel^ mice visited the corners less than WT littermates, which was pronounced during dark phases (shadowed). (**E**,**F**) Ube3a^genedel^ mice exhibited an increase in lick number over the entire period. (**G**,**H**) Control and Ube3a^genedel^ mice exhibited similar learning memory in place preference test. (**I**,**J**) Ube3a^genedel^ mice exhibited less correct visits to rewarding corners than control animals during 10 sessions of reversal learning. Two-way ANOVA with dependent measurements, genotype main effect, *p* < 0.001, depicted with means ± SD. Significant effects of genotype are indicated as * *p* < 0.05, ** *p* < 0.01, *** *p* < 0.001, and **** *p* < 0.0001 for genotype significance.

**Table 1 cells-11-02815-t001:** Summary of Angelman mouse models (previously generated models are summarized and grouped depending on type of mutation).

Group	Strain	Genotype	Phenotype (Ube3a −/+)	Ref.
**Deletions and substitutions**	Ube3a^tm1Alb^	Deletion of 100 N-terminal amino acids in the encoded protein and a frameshift inactivating all putative protein isoforms	Deficits in context-dependent learning (FC), impaired LTP (EP), increased abundance of p53 in PC (IHC), seizures, and motor disfunction (RRT, RCT)	[20]
	Ube3a^tm2Yelg^	Nucleotide substitutions in exon 3 result in a stop codon for glutamic acid at position 113 (E113X).	MNTB neurons decreased failure rate (IJWR), faster recovery after AP in neurons, elevated AP amplitude (IJWR), AIS length increased (IHC), reduced STD and fast recovery from STD (IJWR)	[21]
	Ube3a^em2Yelg^	The ATG codon (exon 3) encoding the start codon methionine of UBE3A isoform 2 was mutated into a TGA, resulting in expression of isoform 3 only.	Cytosolic Ube3a isoform, not critical for development of severe AS symptoms	[18]
	Ube3a^em1Yelg^	An ATG codon (Ube3a exon 4/5) encoding the initiating methionine of UBE3A isoform 3 was mutated into an alanine (GCG). Therefore, only isoform 2 is expressed in these mice.	Nuclear Ube3a isoform, crucial for development. Deficiency leads to synaptic changes and impacts excitation and inhibition balance (VCR). Neurobehavioral phenotype confirmed by RRT NB, MB, FST, OF.	[18]
	Ube3a^em1(IMPC)Hmgu^ (C57BL/6NCrl)	Intra-exon deletion (exon 3)	Assessed by IMPC pipeline, decreased locomotor activity (OF), decreased food intake (IC), decreased respiratory quotient (IC), increased hematocrit (HEM)	[22]
**Floxed alleles**	Ube3a^tm1.1Bdph^	A floxed allele, exon 5 flanked by loxP sites.	Enhanced reward-seeking behavior (OS) due to lack of Ube3a in TH neurons (EP, VCR, IHC)	[23]
	Ube3a^tm1Yelg^	A stop cassette with loxP sites inserted in intron CRE-mediated recombination reinstates gene expression.	Non-CRE recombined mouse recapitulates murine AS phenotype. Recombination leads to partial rescue of the phenotype on the protein (WB), neuronal (EP), and motor behavior levels (OF, MB, RRT, NB, FST)	[24]
	Ube3a^tm1a(KOMP)Wtsi^	The critical exon(s) is/are flanked by loxP sites. FLP recombination generates a conditional allele. Subsequent CRE expression results in a knockout mouse. If CRE expression occurs without FLP expression, a reporter knockout mouse is created.	N/A	[25]
**Inducible isoforms**	Tg(tetO-Ube3a*2)884Svd	The transgene under control of a modified Tet response element (TRE or tetO), transgene of mouse ubiquitin protein ligase E3A (Ube3a) cDNA sequence encoding transcript variant 2 (NM_011668.2) with FLAG tag, and an SV40 polyA signal	Anxiety-like behavior (EPMT, LD) autism (TCSIT), contextual learning deficit (FC), lower seizure threshold (EEG), reduced brain volume (MRI)	[30] unpublisde
	Tg(tetO-Ube3a*1)1Svd	The transgene under control of a modified Tet response element (TRE or tetO), transgene of mouse ubiquitin protein ligase E3A (Ube3a) cDNA sequence encoding transcript variant 1 (NM_173010.3) with FLAG tag, and an SV40 polyA signal	N/A	[31] unpublished
**Modified Ube3a**	Tg(Ube3a)1Mpan	Overexpression of Ube3a gene with three FLAG tags	Overexpression leads to development of autistic symptoms, impaired social behavior (TCSIT, RRT, OF, EMPT), decreased communication and repetitive behavior, impaired glutamatergic synaptic transmission, and glutamate release (EP).	[26]
	Tg(Ube3a)5Mpan	Extra copy of Ube3a transgene in the genome	Increased seizures, impaired social behavior (TCSIT, VT)	[27]
	Ube3a^tm1Jwf^	A part of exon 15 and all of exon 16 are fused to IRES-lacZ-neo cassette, resulting in functional impairment of the C-terminal region responsible for ubiquitin protein ligase activity.	Allows Ube3a expression tracing based on LacZ staining, motor disfunction (RRT, BCT, FST), abnormal EEG), increased abundance of p53 in PC (IHC).	[28,32]
	Ube3a^tm2.1Alb^/Ube3a^tm2Alb^	Fusion of yellow fluorescent protein (YFP) to exon expression of YFP is through inheritance of the maternal allele and recapitulates endogenous expression.	Phenotype not analyzed; the strain is mainly used to track Ube3a expression.	[29]
**Large deletions**	Del(7Gabra3-Ube3a)^1Yhj^	The deletion extending from Gabra3 to Ube3a gene including Atp10a.	Homozygotes exhibit cleft palate and perinatal lethality. In AS mice, impaired behavior (LD, HP, USV, MWM, PPI), motor function (RRT, OF), and seizures	[17]
	Dp(7Herc2-Mkrn3)1Taku	Insertion of selection cassettes and loxP sites proximal to Herc2 and distal to Mkrn3. CRE-mediated recombination in ES cells led to balanced duplication and deletion of 6.3 Mb region between Herc2 and Mkrn3.	Duplication of the paternal allele results in poor social interaction, behavioral inflexibility, and abnormal ultrasonic vocalizations and correlates with anxiety (TCSIT, MWM, BMT, USV, FC) and altered 5-HT2c receptor signaling (EP).	[15]
	Del(7Herc2-Mkrn3)13FRdni	5Mb deletion of entire AS/PWS locus spanning from Herc2 to Mkrn3 genes via Lmp2a transgene insertion	Neurophysiological and behavioral phenotype, cellular morphology, impaired homeostasis and metabolism, increased mortality, aging, and respiratory problems. Analyzed for PWS only.	[19]
	Del(7Ube3a-Snrpn)1Alb	Deletion of genomic DNA from the loxP site within Snrpn to the loxP site within Ube3a	Neurophysiological and behavioral phenotype, impaired growth, increased mortality, aging, and muscle hypotonia. Analyzed for PWS only.	[33]
	Oca2^p−30PUb^	This deletion expands distally from the p locus to Gabrb3, Ube3a, and Ipw. This deletion includes Atp10a.	Used for PWS and Atp10c research, homozygosity lethal, impaired modulation of body fat and/or affecting lipid metabolism (increased total fat)	[34]
	C57BL/6NCrl-Ube3a<em1(IMPC)Ccpcz>/Ph	Gene deletion	Impaired motor functions (RRT, GB, TST) and altered behavior (OF, EPMT, NB, BMT InteliCage)	This article

LTP: long-term potentiation, PC: Purkinje cells, FC: fear conditioning, EP: electrophysiology, TMF: testing motor function, RRT: rotating rod test, BCT: bar-crossing test, IHC: immunohistochemistry, WB: Western blot MNTB: medial nucleus of the trapezoid body, AP: action potential, AIS: axon initial segment, STD: short- term synaptic depression, IJWR: In vivo juxtacellular and whole-cell recordings, VCR: voltage-clamp recording, NB: Nest building, MB: Marble burying, FST: Forced-swim test, OF: Open field test, IC: indirect calorimetry, HEM: hematology, TH: tyrosine-hydroxylase-expressing neurons, OS: Optogenic stimulation, TCSIT: three-chamber social interaction test, EPMT: elevated plus maze tests, VT: video tracking, MWM: Morris water maze, PPI: prepulse inhibition, HP: hot plate, USV: ultrasonic vocalization, LD: light–dark exploration, BMT: Barnes maze test, GB: gait bait, TST: tail suspension test, NOR: novel object recognition, MRI: magnetic resonance imaging.

**Table 2 cells-11-02815-t002:** Sample sizes of animals used for cohorts (number of animals used in groups of every cohort).

Cohort	WT Males	Ube3a^Genedel^ Males	WT Females	Ube3a^Genedel^ Females
1	14	11	10	10
2	10	11	9	10
3	12	12	12	12

## Data Availability

The datasets used and/or analyzed during the current study are available from the corresponding author on reasonable request.

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
