# Peer review of "Generation and Characterization of a Novel Angelman Syndrome Mouse Model with a Full Deletion of the Ube3a Gene"

_cells, 2022, doi:10.3390/cells11182815_

Round 1
Reviewer 1 Report
In this manuscript, the authors provide the behavioral phenotyping of a novel mouse model of Angelman syndrome (AS), obtained through the CRISPR-Cas9 technique. The translational relevance of the model is high and the behavioral assessment performed was appropriate. The major limitations of the proposed model are the lack of anxiety abnormalites and of major cognitive deficits, although additional data analyses are needed to further confirm this result. Indeed, the percent time spent in the open arms, of the elevated plus maze and the percent preference for the novel object in the NOR test should be provided. Overall, the authors have higlighted the strenghts and limitations of their novel animal model quite well. Nonetheless, the text could be improved by minor revisions. In the introduction, the expected impact of sex differences based on human data should be explained. Furthermore, Table 1 should be improved : the phenotypes of the different mouse models of AS should be better described, when possible referring to the specific behavioral tests used. Also, "neurobiological and neurophysiological phenotypes" are too generical terms, that could include also behavioral abnormalities. Concerning the methods, It is mentioned that the NOR test was carried out in a maze, but it is not explained where the objects were located in the maze and whether their relative position was counterbalanced. The authors should also mention whether the estrous cycle of female mice was ever assessed and which impact it could have had on the behavioral results. The discussion should more clearly compare the behavioral results with those from previous mouse models in the same tests and limit as much as possible the speculative interpretations. The impact of sex differences found here should be also discussed, as well as their relevance for the human pathological conditions. The authors should also acknowledge in their discussion the lack of assessment of any autistic-like phenotype and mention this point as a potential follow-up study, if relevant. Electrophysiological or brain measures that could be relevant for assessing the validity of the mouse model in the future should be also clearly mentioned.
Author Response
Dear reviewer,
we thank you for your excellent suggestions that helps improving our paper. Based on you input we have now done the following:
- The english has been edited throughout the manuscript
- The expected impact of sex differences have been addressed in the introduction (lines 59-65)
- Table 1 has been updated
- Method description in NOR has been updated, describing placing of objects (lines 225-228)
- New graph displaying percentage visits in open arms and closed arms in EPM test added (panel 3
- The figure depicting results from NOR displays percentage preference for novel object (time spent with novel object/total time)*100
- More concrete comparison of phenotypes between our model and previously studied ones in discussion and speculative sentences omitted.
- Sex-dependent differences found in the model is discussed and relevance for humans addressed.
- Importance of estrous cycle evaluation mentioned (lines 566-585)
- Phenotypes such as autistic behavior, electrophysiology and EEG measurements were mentioned as important for future studies
Best regards,
the authors
Reviewer 2 Report
The manuscript by Syding et al described a new mouse model Ube3aGenedel and did detailed characterization of this mouse model via a battery of previously published behavior tests. As summarized by the authors, there are a lot of existing mouse models with Ube3a and Angelman syndrome. Many of them were created to address specific hypothesis and shed light to the disease mechanism or therapeutics for AS. Current mouse model created by Syding et al is not too much different from the original AS mouse model with small exonic deletion, therefore I think the scientific novelty is average for this study. However, the entire study was well performed and presented in a clear and detailed way. The results and analysis are scientifically solid.
I do not have other comments.
Author Response
Dear reviewer,
We thank you for taking your time to review our paper and for you feedback.
We have revised the manuscript and done the following, based on the second reviewers suggestions:
- The english has been edited throughout the manuscript
- The expected impact of sex differences have been addressed in the introduction lines 59-65
- Table 1 has been updated
- Method regarding NOR has been updated, describing placing of objects and counterbalancing of relative position lines 225-228
- New graph displaying percentage visits in open arms and closed arms in EPM test added (panel 3)
- Panel 4 updated with boxplot for TST instead of bar graph
- The figure depicting results from NOR displays percentage preference for novel object (time spent with novel object/total time)*100
- More concrete comparison of phenotypes between our model and previously studied ones in discussion land speculative sentences omitted.
- Sex-dependent differences found in the model discussed and relevance for humans addressed.
- Importance of estrous cycle evaluation mentioned lines 566-585
- Phenotypes such as autistic behavior, electrophysiology and EEG measurements were mentioned as important for future studies.
Best regards,
the authors